# Detection of *EGFR* mutations in non-small cell lung cancer by droplet digital PCR

**Drew F. K. Williamson**[1], **Sean R. N. Marris**[1], **Vanesa Rojas-Rudilla**[1], **Jacqueline L. Bruce**[1], **Cloud P. Paweletz**[2], **Geoffrey R. Oxnard**[2], **Lynette M. Sholl**[1], **Fei Dong**[1]*

**1** Department of Pathology, Brigham and Women's Hospital, Boston, MA, United States of America,
**2** Department of Medical Oncology, Dana Farber Cancer Institute, Boston, MA, United States of America

* fdong1@bwh.harvard.edu

**Data Availability Statement:** All relevant data are within the manuscript and its Supporting Information files.

**Funding:** The authors received no specific funding for this work.

## Abstract

Activating mutations in *EGFR* predict benefit from tyrosine kinase inhibitor therapy for patients with advanced non-small cell lung cancer. Directing patients to appropriate therapy depends on accurate and timely *EGFR* assessment in the molecular pathology laboratory. This article describes the analytical design, performance characteristics, and clinical implementation of an assay for the rapid detection of *EGFR* L858R and exon 19 deletion mutations. A droplet digital polymerase chain reaction (ddPCR) assay was implemented with probe hydrolysis-dependent signal detection. A mutation-specific probe was used to detect *EGFR* L858R. A loss of signal design was used to detect *EGFR* exon 19 deletion mutations. Analytical sensitivity was dependent on DNA input and was as low as 0.01% variant allele fraction for the *EGFR* L858R assay and 0.1% variant allele fraction for the *EGFR* exon 19 deletion assay. Correlation of 20 clinical specimens tested by ddPCR and next generation sequencing showed 100% concordance. ddPCR showed 53% clinical sensitivity in the detection of *EGFR* mutations in plasma cell-free DNA from patients with lung cancer. The median clinical turnaround time was 5 days for ddPCR compared to 13 days for next generation sequencing. The findings show that ddPCR is an accurate and rapid method for detecting *EGFR* mutations in patients with non-small cell lung cancer.

## Introduction

Lung cancer remains the second most common non-skin cancer of both men and women, and the most common overall, with an estimated 229,000 Americans diagnosed in 2020 [1]. Approximately 20% of patients with non-small cell lung cancer have mutations in the *EGFR* gene [2, 3], for which targeted tyrosine kinase inhibitor therapy provides significant treatment benefit [4].

Given the importance of matching patients with *EGFR* mutations to the appropriate targeted therapy, laboratories require testing approaches that are both timely and accurate. The most common mutations in *EGFR* that confer sensitivity are *EGFR* L858R (a point mutation in exon 21) and in frame deletions in exon 19. The gold standard for detecting these mutations has historically been *EGFR* sequencing, often by Sanger sequencing performed on formalin-

**Competing interests:** I have read the journal's policy and the authors of this manuscript have the following competing interests: Geoffrey Oxnard is an employee of Foundation Medicine and holds equity in Roche. Lynette Sholl is a consultant for Genentech and Eli Lilly and receives research funding from Genentech. We can confirm that Dr. Oxnard's and Dr. Sholl's affiliations do not alter our adherence to PLOS ONE policies on sharing data and materials.

fixed paraffin-embedded (FFPE) tissue [5, 6]. However, the availability of next generation sequencing (NGS) and new polymerase chain rection (PCR) technologies has opened the possibility for different approaches.

A recent technology that enables rapid mutational testing is droplet digital PCR (ddPCR). This assay is a modification of PCR wherein template molecules are distributed into many parallel reactions at limiting dilutions and each run to completion [7]. Using microfluidics, independent PCR reactions are suspended in aqueous droplets in oil, and sensitive optical systems can detect fluorescence from individual droplets in single or multiple channels [8, 9]. This setup provides a highly sensitive and specific system that is robust enough for use in the clinical laboratory, and such systems have been shown to be useful for a range of applications, from quantitation of DNA copy number [10] to analysis of circulating tumor DNA [11] and detection of SARS-CoV-2 [12].

Here we present our work validating a dual channel ddPCR system for *EGFR* L858R and exon 19 deletion detection in plasma and FFPE tissue. We describe the analytical validation for each assay, and we demonstrate that the assay has excellent performance characteristics and superior turnaround time compared to panel next generation sequencing.

## Materials and methods

### DNA isolation

DNA was isolated from formalin-fixed, paraffin-embedded tumor tissue. Tumor-enriched areas were macrodissected from ten 4-μm sections. DNA was isolated using QIAamp DNA mini kit (Qiagen; Hilden, Germany) and quantified using Qubit-based dsDNA detection (Qubit 3.0, Life Technologies; Carlsbad, CA). Cell-free genomic DNA (cfDNA) was isolated from plasma using the QIAamp Circulating Nucleic Acid Kit and then quantified using Qubit-based dsDNA detection. DNA was then diluted to a maximum concentration of 3 ng/μL.

### Droplet digital PCR

DNA was mixed with commercially available master mix (Bio-Rad; Hercules, CA), PCR primers and fluorescently labeled probes (Applied Biosystems; Foster City, CA) for detection of the respective sequences (**Table 1**). A total volume of 20 μL was used in each reaction, comprised of 12.5 μL master mix, 0.625 μL of 40X Taqman probes, and 6.875 μL of RNase/DNase-free water, and 5 μL of sample DNA or control. For clinical testing, control samples were included with each run: water, a wild type control, as well as low (2% mutant DNA) and high (20% mutant DNA) controls for each mutation. Aqueous droplets containing individual PCR templates were generated in a water-oil emulsion-based droplet generator using 70 μL of oil per reaction. Using 50 μL of the resulting droplets, PCR for 40 cycles of alternating 94˚C and 58˚C was performed for detection of the respective sequences. These droplets were analyzed by flow cytometry (Bio-Rad QX200 ddPCR System), which detected probe-specific fluorescent signals. Flow cytometry event counts were used to detect wild type and mutant signals and quantify variant allele fraction.

### Description of assay design

Independent ddPCR reactions were designed to detect the presence of *EGFR* c.2573T>G (L858R) and *EGFR* exon 19 deletion (**Fig 1**).

For *EGFR* L858R detection, the assay contained a single set of primers and two competitive probes, one detecting the wild-type allele (VIC) and one detecting the mutant allele (FAM).

**Table 1. Primer and probe sequences for *EGFR* mutation detection.**

| | *EGFR* L858R | *EGFR* exon 19 deletion |
|---|---|---|
| Forward Primer | GCAGCATGTCAAGATCACAGATT | GTGAGAAAGTTAAAATTCCCGTC |
| Reverse Primer | CCTCCTTCTGCATGGTATTCTTTCT | CACACAGCAAAGCAGAAAC |
| Wild Type Probe | AGTTTGGCCAGCCCAA | ATCGAGGATTTCCTTGTTG |
| Mutant Probe | AGTTTGGCCCGCCCAA | AGGAATTAAGAGAAGCAACATC |

Wild type events in the VIC channel ensured that *EGFR* DNA was present in the sample. Mutant events were represented as positive events in the FAM channel.

For exon 19 deletion mutations, a VIC-labeled probe hybridized to a shared reference sequence in exon 19, while a FAM-labeled probe only hybridized to the sequence spanning the deletion hotspot. Droplets with wild type DNA showed positive signal for both VIC and FAM. Droplets with deletion showed VIC signal only (**Fig 2**).

## Control *EGFR* experiment

Control *EGFR* L858R, exon 19 deletion (*EGFR* p.E746_A750del), and wild type DNA was obtained from Horizon Discovery (Horizon Discovery Ltd., Waterbeach, United Kingdom). Serial dilutions of mutant DNA were performed to generate DNA at 10%, 1%, 0.1%, 0.01%, and 0% variant allele fraction. Total *EGFR* DNA concentration was loaded at 250, 125, 62.5, 31.25 and 15.625 ng. Each combination of variant allele fraction and DNA concentration was run in duplicate.

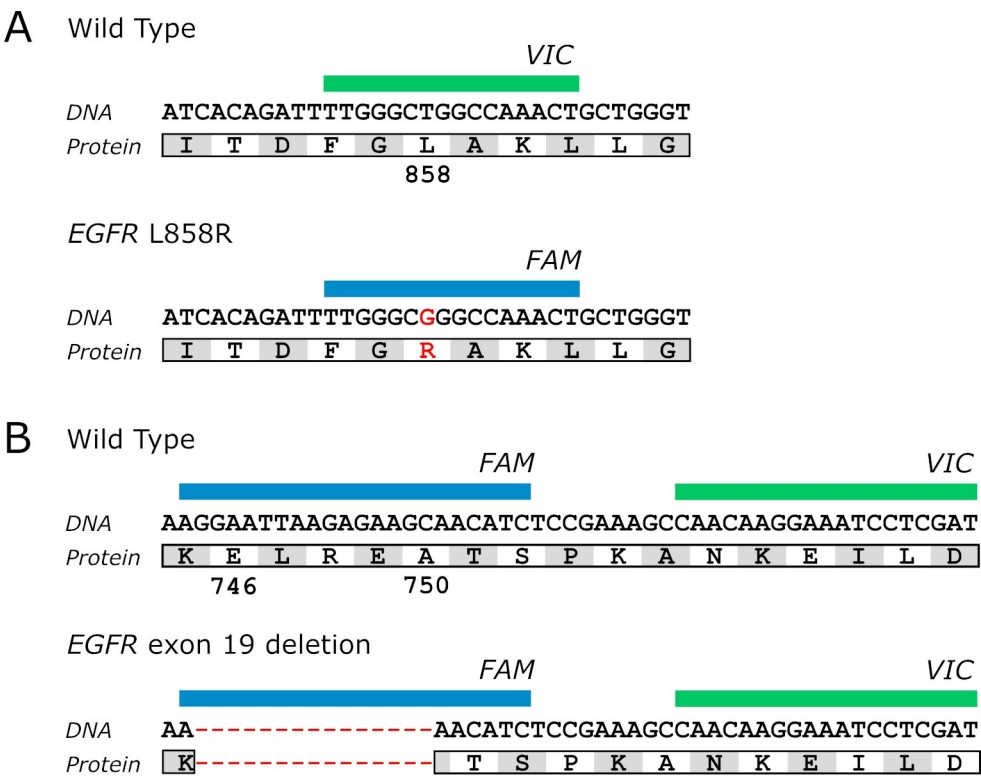

**Fig 1. Droplet digital PCR probe design to detect *EGFR* L858R substitution (A) and *EGFR* exon 19 deletion (B).**

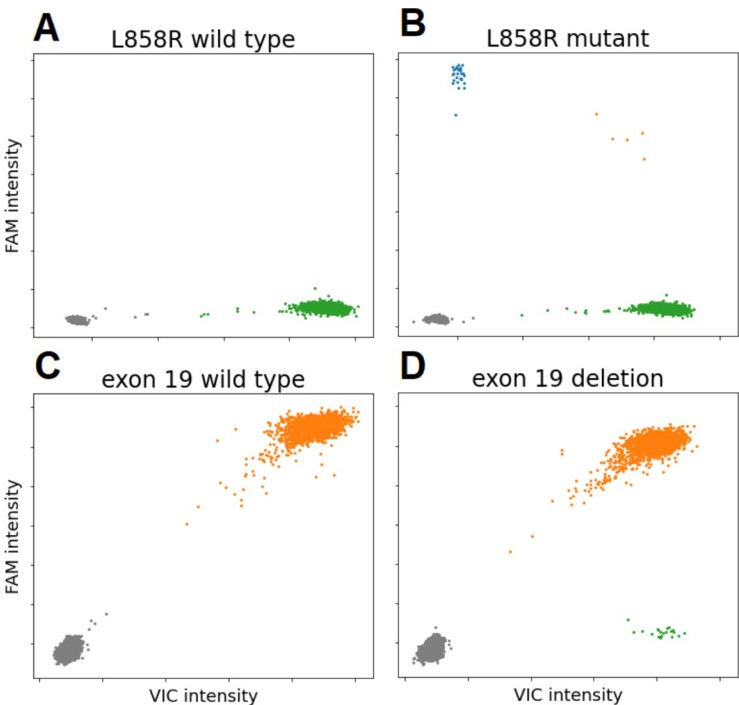

**Fig 2.** Droplet digital PCR test for *EGFR* L858R for wild type DNA (A) and mutant *EGFR* L858R DNA at 1% variant allele fraction (B). *EGFR* exon 19 deletion analysis for wild type DNA (C) and mutant *EGFR* E746_A750del at 1% variant allele fraction (D).

## DNA quantitation

DNA concentration could be calculated based on a Poisson distribution,

$$\lambda = -\ln(1 - p)$$

where $\lambda$ represented number of target DNA molecules and $p$ was the ratio of positive events to all events. The principles of DNA quantitation in digital PCR systems have been previously described [13]. Both mutant DNA concentration and wild type DNA concentration could be derived to calculate the mutation variant allele fraction.

## OncoPanel next generation sequencing

Orthogonal testing was performed with OncoPanel, a targeted next generation sequencing panel, as previously described [14]. Briefly, DNA was isolated from formalin-fixed, paraffin-embedded tissue with at least 20% tumor purity. Hybrid capture was performed to enrich for the coding regions of 447 cancer associated genes including *EGFR* using the Agilent SureSelect XT Fast Reagent Kit (Agilent Technologies Inc., Santa Clara, CA). Sequencing was performed on an Illumina HiSeq 2500 sequencer (Illumina Inc., San Diego, CA). Informatics was performed with a custom pipeline.

On validation, OncoPanel demonstrated 97.8% sensitivity and 100% specificity for single nucleotide variants and 100% sensitivity and 100% specificity for insertion and deletion variants excluding *FLT3* and *NPM1* variants. The limit of detection was set at 10% variant allele fraction for regions of at least 50x read depth, achieving 98.4% concordance in triplicate runs for variants that meet these criteria [14]. For clinical lung cancer specimens, *EGFR* mutational hotspots in exon 19 and exon 21 underwent additional manual inspection in Integrative Genomics Viewer.

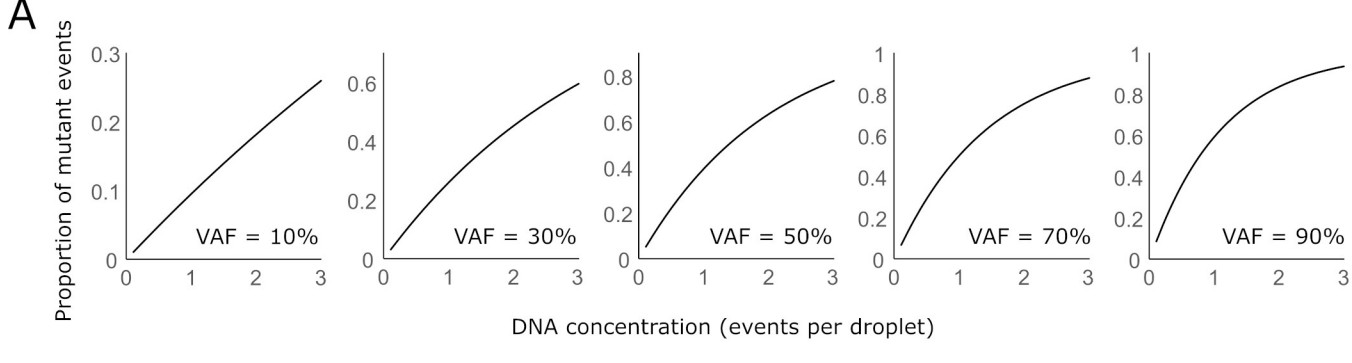

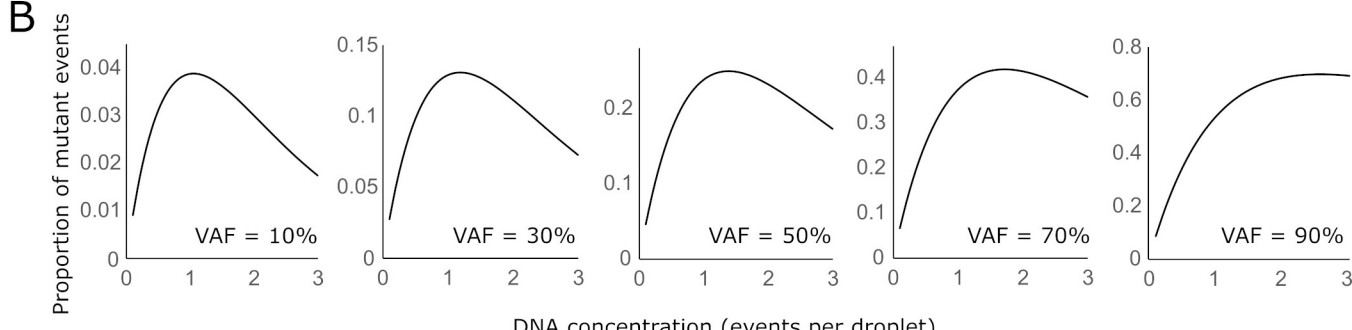

**Fig 3.** Relationship between variant allele fraction (VAF), DNA loading concentration, and positive mutation events for *EGFR* L858R (A) and *EGFR* exon 19 deletion (B).

### Patient selection

Patient specimens tested by ddPCR and OncoPanel were identified by retrospective review of the medical record. For patients who had multiple cfDNA specimens tested by ddPCR, the result from the earliest specimen was used for analysis. This study was approved by the Mass General Brigham Human Research Committee. As the research was deemed no more than minimal risk, patient consent was waived by the ethics committee.

## Results

### Calculated effect of loading concentration on mutation detection

We calculated how DNA loading affected performance characteristics for the *EGFR* L858R and exon 19 deletion ddPCR assay designs. For the *EGFR* L858R assay, positive mutation events increased as DNA concentration increased (**Fig 3A**). In specimens with low variant allele fraction, the relationship of positive events increased linearly with DNA concentration.

For the *EGFR* exon 19 deletion assay, VIC-only positive events increased with DNA concentration at low concentration. As the total DNA concentration increased past the optimal loading concentration, the proportion of positive events began to decrease (**Fig 3B**). As mutant variant allele fraction approached 0, the optimal loading concentration approached 1 event per droplet.

### Analytical specificity

Wild type *EGFR* was tested across five DNA loading concentrations (250, 125, 62.5, 31.25, and 15.625 ng). Each concentration was run in duplicate for both the L858R and exon 19 deletion

ddPCR assays. The 10 *EGFR* L858R wild type reactions generated a median of 15,282 droplets (range 7,954 to 18,141 droplets). Five reactions showed a single false positive event, and no reaction showed greater than one false positive event. The overall per droplet false positive error rate was 0.003% for the L858R reaction. In 10 *EGFR* exon 19 deletion reactions with wild type *EGFR* control DNA, no false positive mutant events were observed. Based on this observation, the assay was interpreted to be positive for *EGFR* mutation if 3 or more mutant events were present. With a positive threshold set at 3 events in duplicate reactions, the expected false positive rate in *EGFR* L858R analysis was 0.02%.

## DNA quantification and limit of detection

Quantitation of DNA concentration based on Poisson approximation showed that calculated DNA concentration was linear with respect to DNA input for both the L858R and exon 19 deletion reactions (**Fig 4**, $R^2 = 0.986$ and $R^2 = 0.998$, respectively).

The limit of detection depended on input DNA concentration. Reactions were performed with mutant L858R or exon 19 deletion DNA at 0.01%, 0.1%, 1%, and 10% variant allele fraction across five DNA loading concentrations (250, 125, 62.5, 31.25, and 15.625 ng). Each reaction was performed in duplicate. A positive result was defined as generating at least 3 positive droplet events in both replicate reactions.

For the L858R reaction, the limit of detection was 0.01% variant allele fraction at 250 ng DNA input, 0.1% variant allele fraction at 125 and 62.5 ng input, and 1% at 31.25 and 15.625 ng input (**Table 2**). For the exon 19 deletion reaction, the limit of detection was 0.1% variant allele fraction at 250, 125, and 62.5 ng DNA input and 1% at 31.25 and 15.625 ng input (**Table 3**).

The observed difference in limit of detection between the L858R and exon 19 deletion assays was expected based on assay design principles. The analytical sensitivity decreased for the exon 19 deletion reaction at the high DNA concentration (see "Calculated Effect of Loading Concentration on Mutation Detection").

## Accuracy and precision

For reactions above the limit of detection, a comparison of calculated variant allele fraction to true variant allele fraction was shown to be linear for both the L858R and exon 19 deletion reactions with $R^2 = 0.998$ and 0.995, respectively.

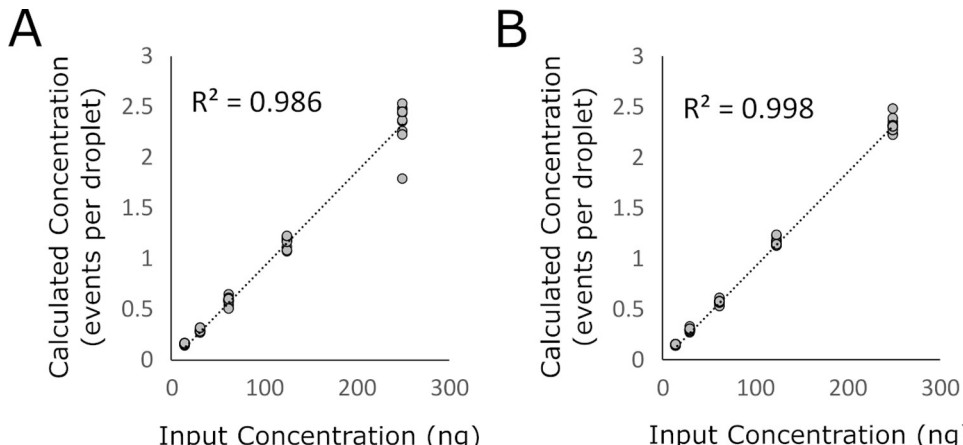

**Fig 4.** Linear correlation between input DNA concentration and experimentally calculated concentration for *EGFR* L858R reaction (A) and *EGFR* exon 19 deletion reaction (B).

**Table 2. Number of positive mutant events for *EGFR* L858R reactions.**

| | | Input Concentration (ng) | | | | |
|---|---|---|---|---|---|---|
| | | 15.625 | 31.25 | 62.5 | 125 | 250 |
| Variant Allele Fraction | 0% | 0 | 1 | 0 | 1 | 1 |
| | | 0 | 0 | 1 | 0 | 1 |
| | 0.01% | 0 | 1 | 3 | 4 | 4 |
| | | 0 | 2 | 1 | 2 | 6 |
| | 0.1% | 0 | 2 | 11 | 17 | 39 |
| | | 3 | 6 | 12 | 22 | 32 |
| | 1% | 21 | 43 | 119 | 209 | 428 |
| | | 31 | 33 | 88 | 219 | 330 |
| | 10% | 267 | 592 | 1146 | 1951 | 2504 |
| | | 249 | 503 | 422 | 1733 | 3426 |

Each combination of DNA input concentration and variant allele fraction is performed in duplicate. Positive results with at least three mutant events in each replicate are highlighted in gray.

Accuracy and precision of quantitation improved with increasing mutant events and variant allele fraction. For the L858R reaction, average percent error per reaction was 32.4%, 15.7%, 11.0% and 2.6% at variant allele fractions of 0.01%, 0.1%, 1%, and 10%, respectively. The coefficient of variation was 0.143, 0.172, 0.141 and 0.036.

For the exon 19 deletion reaction, average percent error per reaction was 53.7%, 11.7%, and 4.1% at variant allele fractions of 0.1%, 1%, and 10%, respectively. The coefficient of variation was 0.425, 0.131 and 0.056 (**Fig 5**).

## Orthogonal validation and clinical sensitivity

We reviewed the application of the *EGFR* ddPCR assay in our clinical practice. Twenty patients had both ddPCR and OncoPanel targeted next generation sequencing performed on tissue specimens on the same pathological specimen. One cancer had *EGFR* L858R, five had *EGFR* exon 19 deletion, and 14 were wild type for *EGFR*. Overall concordance between ddPCR and OncoPanel in this cohort was 100%.

**Table 3. Number of positive mutant events for *EGFR* exon 19 deletion reactions.**

| | | Input Concentration (ng) | | | | |
|---|---|---|---|---|---|---|
| | | 15.625 | 31.25 | 62.5 | 125 | 250 |
| Variant Allele Fraction | 0% | 0 | 0 | 0 | 0 | 0 |
| | | 0 | 0 | 0 | 0 | 0 |
| | 0.01% | 0 | 0 | 1 | 1 | 0 |
| | | 0 | 0 | 1 | 0 | 0 |
| | 0.1% | 2 | 1 | 10 | 5 | 4 |
| | | 2 | 9 | 5 | 13 | 9 |
| | 1% | 28 | 36 | 42 | 72 | 43 |
| | | 21 | 51 | 65 | 70 | 40 |
| | 10% | 198 | 359 | 630 | 627 | 424 |
| | | 245 | 412 | 553 | 643 | 434 |

Each combination of DNA input concentration and variant allele fraction is performed in duplicate. Positive results with at least three mutant events in each replicate are highlighted in gray.

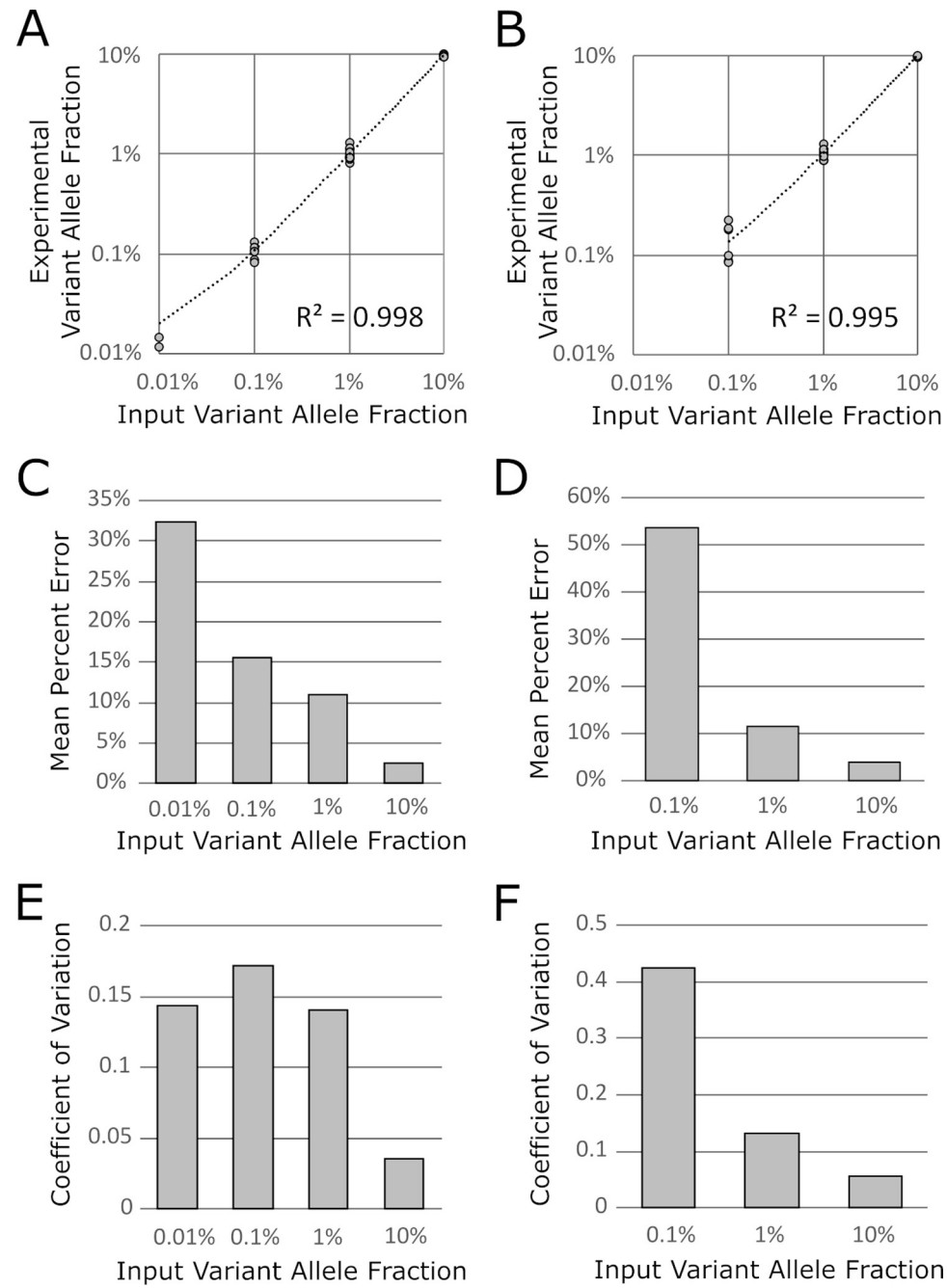

**Fig 5.** Experimental versus input variant allele fraction for specimens above limit of detection for *EGFR* L858R (A) and *EGFR* exon 19 deletion (B). Mean percent error decreases with increasing variant allele fraction for *EGFR* L858R (C) and *EGFR* exon 19 deletion (D). Coefficient of variation with respect to variant allele fraction for *EGFR* L858R (E) and *EGFR* exon 19 deletion (F).

The *EGFR* ddPCR assay was also used to detect circulating cell free tumor DNA from patients' plasma specimens. In total, 183 patients with advanced lung cancer had ddPCR performed on plasma as well as tissue evaluated by OncoPanel sequencing. Of 92 cancers with *EGFR* L858R or exon 19 deletion mutation detected by OncoPanel, 49 had mutations identified by ddPCR in plasma specimens. The overall clinical sensitivity of plasma cell free

mutational analysis was 53%. An additional four plasma specimens from patients with *EGFR*-mutated lung cancers showed mutant events in replicate reactions but did not meet the criteria for reporting (at least 3 events in each replicate reaction). No false positive results were observed.

In this cohort of patients, the median age was 64 years (range 31 to 97 years) with 64 (70%) female patients. New diagnoses accounted for 28 of 92 (30%) patients with *EGFR*-mutated cancers with 82 (89%) being at stage IV and 89 (97%) being at stage III or IV. Most patients had received some form of treatment prior to their first plasma ddPCR assay, with 52 (57%) receiving a tyrosine kinase inhibitor, 21 (23%) receiving chemotherapy, and 36 (39%) receiving localized therapy to their primary tumor either in the form of surgery (20, 22%) or radiotherapy (16, 17%). Only 28 (30%) had received no pre-testing therapy for their primary tumor at all.

Sensitivity among patients with Stage IV disease was 59%, while for patients with Stage I-II disease, it was only 17%. Sensitivity also varied by treatment received, with sensitivity for patients receiving no treatment being 71%, for those receiving a tyrosine kinase inhibitor being 44%, and for those receiving any systemic therapy (tyrosine kinase inhibitor, chemotherapy, or checkpoint inhibitor) being 37%.

## Turnaround time

Finally, we evaluated assay turnaround time, the number of days from when the specimen arrived in the laboratory to when the pathology report was generated, for both ddPCR and next generation sequencing in patients who had both assays performed. The median turnaround time for OncoPanel sequencing was 13 days (range 6 to 133 days). The median turnaround time for ddPCR was 6 days (range 1 to 10 days) for tissue specimens and 4 days (range 1 to 11 days) for plasma specimens. The difference in turnaround time between next generation sequencing and ddPCR was statistically significant (**Fig 6**, p < 0.001).

## Discussion

Here, we report the results of our validation for using a dual channel ddPCR system for detection of *EGFR* L858R and exon 19 deletion in FFPE and plasma. Our laboratory detects *EGFR* exon 19 deletions via a loss-of-signal testing strategy. Both loss-of-signal and gain-of-signal approaches have been previously described [15, 16]. The loss-of-signal approach allows for detection of a range of mutations without *a priori* knowledge of exact deletion breakpoints in exon 19. Concordance between NGS and ddPCR performed on the same tumor specimen was 100%, demonstrating that ddPCR is an accurate method for rapid assessment of *EGFR* mutations.

The low limit of detection for ddPCR makes the technology ideal for use in mutation detection in plasma that may be undetectable by other molecular genotyping methods. In our cohort, ddPCR performed on plasma cell free tumor DNA has a sensitivity of 53% with 100% specificity. The clinical sensitivity is lower compared to that reported in prior published studies which range from 63% to 82% [15–18]. The difference may reflect patient referral patterns in our practice and analytical limitations in clinical specimens. Our cohort has a slightly lower proportion of stage IV patients than most of those previously published. Additionally, more than half of our patients (56%) had already been treated with one or more TKIs at the time of testing which may reduce the ability of our assay to detect mutations, as evidenced by the markedly decreased sensitivity of our assay for patients who has received systemic therapy (37%) versus those who had not received any treatment at all (71%).

As NGS panels have become increasingly utilized for molecular characterization of advanced lung cancers, attempts have been made to find optimal testing strategies. NGS, while

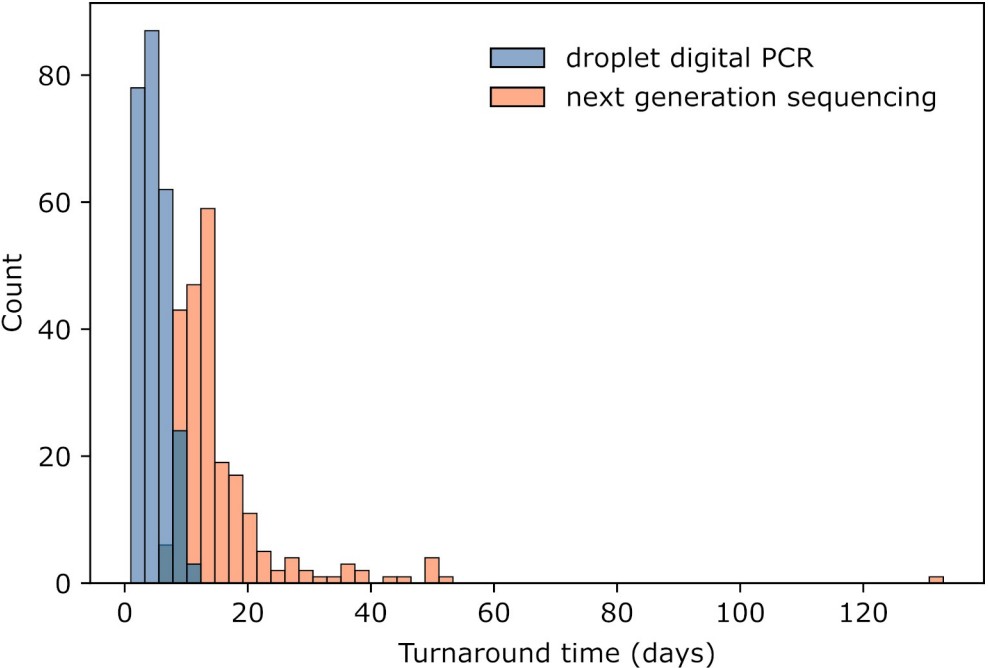

**Fig 6. Histogram of turnaround times for droplet digital PCR compared to panel next generation sequencing.**

accurate and comprehensive, may take weeks to perform, with some patients with actionable mutations starting first-line chemotherapy before results are available [19]. Patients whose tumors harbor targetable mutations may benefit from rapid tests that are more limited in scope [20]. In our institutional experience, the median turnaround time for ddPCR is seven days faster than NGS. Early identification of actional *EGFR* alterations may aid in the timely selection of targeted therapy for patients who are acutely ill.

Plasma ddPCR can also be used as a tool for monitoring treatment response and tumor evolution. Historically, ddPCR has been used to monitor the *EGFR* T790M mutation, which confers resistance to early generation tyrosine kinase inhibitors erlotinib and gefitinib. Several studies have shown the utility of ddPCR for detecting this mutation over time [21–24]. This diagnostic testing strategy will have to be modified as osimertinib becomes a first line modality for *EGFR*-mutated lung cancers [25], leading to different mechanisms of therapeutic resistance [26].

## Conclusions

This article describes our validation and clinical experience with a clinical ddPCR assay to test for common *EGFR* mutations. Experimentation confirms the analytical principles of DNA quantification by ddPCR. Clinical *EGFR* analysis by ddPCR is shown to be accurate with an added benefit of faster turnaround time compared to panel next generation sequencing. ddPCR is an accurate and rapid method for the detection of *EGFR* mutations with clinical utility for patients with advanced lung cancer.

## Supporting information

**S1 Dataset.**
(DOCX)

## Author Contributions

**Conceptualization:** Fei Dong.

**Data curation:** Drew F. K. Williamson, Sean R. N. Marris.

**Formal analysis:** Drew F. K. Williamson, Fei Dong.

**Investigation:** Drew F. K. Williamson, Sean R. N. Marris, Vanesa Rojas-Rudilla, Jacqueline L. Bruce, Cloud P. Paweletz, Geoffrey R. Oxnard, Lynette M. Sholl, Fei Dong.

**Methodology:** Cloud P. Paweletz, Geoffrey R. Oxnard, Lynette M. Sholl, Fei Dong.

**Supervision:** Vanesa Rojas-Rudilla, Jacqueline L. Bruce, Fei Dong.

**Writing – original draft:** Drew F. K. Williamson, Fei Dong.

**Writing – review & editing:** Drew F. K. Williamson, Sean R. N. Marris, Vanesa Rojas-Rudilla, Jacqueline L. Bruce, Cloud P. Paweletz, Geoffrey R. Oxnard, Lynette M. Sholl, Fei Dong.

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
