## [Decision Letter · Decision Letter 0]

31 Aug 2021

PONE-D-21-21322

Detection of EGFR mutations in non-small cell lung cancer by droplet digital PCR

PLOS ONE

Dear Dr. Dong,

Thank you for submitting your manuscript to PLOS ONE. After careful consideration, we feel that it has merit but does not fully meet PLOS ONE’s publication criteria as it currently stands. Therefore, we invite you to submit a revised version of the manuscript that addresses the points raised during the review process.

We look forward to receiving your revised manuscript.

Kind regards,

Alvaro Galli

Academic Editor

PLOS ONE

2. Please provide additional details regarding participant consent. In the Methods section, please ensure that you have specified (1) whether consent was informed and (2) what type you obtained (for instance, written or verbal). If your study included minors, state whether you obtained consent from parents or guardians. If the need for consent was waived by the ethics committee, please include this information.

3. Thank you for providing the following Funding Statement:

“The authors received no specific funding for this work.”

We note that one or more of the authors is affiliated with the funding organization, indicating the funder may have had some role in the design, data collection, analysis or preparation of your manuscript for publication; in other words, the funder played an indirect role through the participation of the co-authors.

If the funding organization did not play a role in the study design, data collection and analysis, decision to publish, or preparation of the manuscript and only provided financial support in the form of authors' salaries and/or research materials, please review your statements relating to the author contributions, and ensure you have specifically and accurately indicated the role(s) that these authors had in your study in the Author Contributions section of the online submission form. Please make any necessary amendments directly within this section of the online submission form. Please also update your Funding Statement to include the following statement: “The funder provided support in the form of salaries for authors [insert relevant initials], but did not have any additional role in the study design, data collection and analysis, decision to publish, or preparation of the manuscript. The specific roles of these authors are articulated in the ‘author contributions’ section.”

If the funding organization did have an additional role, please state and explain that role within your Funding Statement.

Please also provide an updated Competing Interests Statement declaring this commercial affiliation along with any other relevant declarations relating to employment, consultancy, patents, products in development, or marketed products, etc.

Within your Competing Interests Statement, please confirm that this commercial affiliation does not alter your adherence to all PLOS ONE policies on sharing data and materials by including the following statement: "This does not alter our adherence to PLOS ONE policies on sharing data and materials.” (as detailed online in our guide for authors http://journals.plos.org/plosone/s/competing-interests). If this adherence statement is not accurate and there are restrictions on sharing of data and/or materials, please state these. Please note that we cannot proceed with consideration of your article until this information has been declared

“I have read the journal's policy and the authors of this manuscript have the following competing interests:

Geoffrey Oxnard is an employee of Foundation Medicine and holds equity in Roche. Lynette Sholl is a consultant for Genentech and Eli Lilly and receives research funding from Genentech.”

Reviewers' comments:

Reviewer's Responses to Questions

**Comments to the Author**

1. Is the manuscript technically sound, and do the data support the conclusions?

Reviewer #1: Partly

Reviewer #2: Partly

2. Has the statistical analysis been performed appropriately and rigorously? 

Reviewer #1: I Don't Know

Reviewer #2: Yes

3. Have the authors made all data underlying the findings in their manuscript fully available?

Reviewer #1: No

Reviewer #2: Yes

4. Is the manuscript presented in an intelligible fashion and written in standard English?

Reviewer #1: Yes

Reviewer #2: Yes

5. Review Comments to the Author

Reviewer #1: 1, the study was to generate more data to demonstrate potential value of ddPCR for clinical use in detecting tumor mutations especially with ctDNA. The manuscript is in good writen English.

2, more detailed technical information should be provided for readers to better understanding of the potential value and limitation of the ddPCR assays, including 1) brief procedure of the assays; 2) cfDNA/plasma input for the clinical validation; 3) key experimental and performance parameters of the reference NGS-panel assay, such as LOD, sensitivity, specificity.

3, there is a significant difference in terms of technical sensitivity or LOD between the ddPCR assay for L858R and that for E19 deletions, which needs an explanation or interpretation.

4, in the false positive tests, a few reactions reported single false positive event for L858R, and no false posive for E19 deletions, but the positive calling cutoff were set to be 3 positive droplets, which needs an explanation of rational. Considering the value of such assays to to detect ctDNA mutations which is at very limited concentration, maximization of clinical sensitivity with good control of specificity is important. 2 positive droplets from a single reaction or even duplicated reactions might be better.

5, there is a significant difference in terms of analytical and clinical sensitivity between the ddPCR assays and that reported previously such as ref. 15. This should be discussed with sufficiently deepth so that the readers can understand the reasons behind.

6, considering the broad application of NGS for ctDNA in clinic, it would be more informative if there is a head-to-head comparison between the ddPCR assays and a NGS assay for ctDNA detection.

Reviewer #2: The manuscript is aimed at validating a dual channel ddPCR system for EGFR L858R and exon 19 deletion

detection in plasma and FFPE tissue.

Despite the analysis have been conducted in a very rigorous way, the number of mutant samples analysed on FFPE is very low (6 out of 20) and the advantages in terms of specificity and turn around time of a ddPCR are well known.

Considering the low sensitivity (only 52%) using plasma samples, I would suggest the authors to accurately consider/discuss the clinical characteristics of patients, since this is one of the major challenge of liquid biopsy. Patients should try to set up their assay on an homogeneous population, to cut out any possible analytical bias to confirm their assay works.

6. PLOS authors have the option to publish the peer review history of their article (what does this mean?). If published, this will include your full peer review and any attached files.

Reviewer #1: **Yes: **Guanshan Zhu

Reviewer #2: No

---

## [Author Response · Author response to Decision Letter 0]

28 Jan 2022

Dear reviewers,

Thank you for taking time to review our article and for your helpful comments. To comply with PLOS data availability guidelines, we have provided the minimal data set for all study results in a Supplemental File.

Please see the full response to each point of your suggestions below:

Reviewer #1:

1, the study was to generate more data to demonstrate potential value of ddPCR for clinical use in detecting tumor mutations especially with ctDNA. The manuscript is in good writen English.

Thank you for your comment.

2, more detailed technical information should be provided for readers to better understanding of the potential value and limitation of the ddPCR assays, including 1) brief procedure of the assays; 2) cfDNA/plasma input for the clinical validation; 3) key experimental and performance parameters of the reference NGS-panel assay, such as LOD, sensitivity, specificity.

We have updated methods to be more comprehensive. We have expanded the methods section to include more information from our protocol. We have specified the plasma input for cfDNA analysis. The method section for DNA isolation and ddPCR now reads as follows:

DNA Isolation

DNA was isolated from formalin-fixed, paraffin-embedded tumor tissue. Tumor-enriched areas were macrodissected from ten 4-μm sections. DNA was isolated using QIAamp DNA mini kit (Qiagen; Hilden, Germany) and quantified using Qubit-based dsDNA detection (Qubit 3.0, Life Technologies; Carlsbad, CA). Cell-free genomic DNA (cfDNA) was isolated from plasma using the QIAamp Circulating Nucleic Acid Kit and then quantified using Qubit-based dsDNA detection. DNA was then diluted to a maximum concentration of 3 ng/μL.

Droplet Digital PCR

DNA was mixed with commercially available master mix (Bio-Rad; Hercules, CA), PCR primers and fluorescently labeled probes (Applied Biosystems; Foster City, CA) for detection of the respective sequences (Table 1). A total volume of 20 μL was used in each reaction, comprised of 12.5 μL master mix, 0.625 μL of 40X Taqman probes, and 6.875 μL of RNase/DNase-free water, and 5 μL of sample DNA or control. For clinical testing, control samples were included with each run: water, a wild type control, as well as low (2% mutant DNA) and high (20% mutant DNA) controls for each mutation. Aqueous droplets containing individual PCR templates were generated in a water-oil emulsion-based droplet generator using 70 μL of oil per reaction. Using 50 μL of the resulting droplets, PCR for 40 cycles of alternating 94 °C and 58 °C was performed for detection of the respective sequences. These droplets were analyzed by flow cytometry (Bio-Rad QX200 ddPCR System), which detected probe-specific fluorescent signals. Flow cytometry event counts were used to detect wild type and mutant signals and quantify variant allele fraction.

We have added more details about the NGS panel assay validation, including LOD, sensitivity, and specificity. The following text has been added to the manuscript:

On validation, OncoPanel demonstrated 97.8% sensitivity and 100% specificity for single nucleotide variants and 100% sensitivity and 100% specificity for insertion and deletion variants excluding FLT3 and NPM1 variants. The limit of detection was set at 10% variant allele fraction for regions of at least 50x read depth, achieving 98.4% concordance in triplicate runs for variants that meet these criteria.[14] 

3, there is a significant difference in terms of technical sensitivity or LOD between the ddPCR assay for L858R and that for E19 deletions, which needs an explanation or interpretation.

Thank you for your question. The difference in technical sensitivity/LOD between the L858R and E19 deletion assays is due to assay design (Figure 1 and Figure 2) and has to do with expected results as DNA concentration increases beyond limiting dilution. In the L858R assay, a droplet containing both wild type and mutant alleles will be double-positive for VIC and FAM and will contribute to the calculation for both wild type and mutant alleles. In the E19del assay, a droplet containing both wild type and mutant alleles will also be double-positive for VIC and FAM and will be indistinguishable from wild type only droplets. The end effect is that for the E19del assay, the analytical sensitivity decreases at high loading concentrations and is evident in the experimental data in Table 3.

This concept is explored in the section "Calculated Effect of Loading Concentration on Mutation Detection", including in Figure 3.

We have added the following text to the relevant section ("DNA Quantification and Limit of Detection") to help clarify this for the reader:

The observed difference in limit of detection between the L858R and exon 19 deletion assays was expected based on assay design principles. The analytical sensitivity decreased for the exon 19 deletion reaction at the highest DNA concentration (see "Calculated Effect of Loading Concentration on Mutation Detection").

4, in the false positive tests, a few reactions reported single false positive event for L858R, and no false posive for E19 deletions, but the positive calling cutoff were set to be 3 positive droplets, which needs an explanation of rational. Considering the value of such assays to to detect ctDNA mutations which is at very limited concentration, maximization of clinical sensitivity with good control of specificity is important. 2 positive droplets from a single reaction or even duplicated reactions might be better.

This is an interesting question about how to set thresholds for reporting based on data in analytical validation.

For the L858R assay, the overall per-event false positive rate is very low: 5/145019 = 0.003% of any droplet may show a false positive result. However, this risk is cumulative over 15,000 droplets per reaction. The expected number of false positive events is 15,000 x 0.003% = 0.517. This is consistent with the observed data, where 5/10 L858R reactions showed one false positive event each.

This data can be further modeled by a Poisson distribution with *λ* = 0.517. The probability of two or more positive event in a single reaction is 0.0955, and the probability of two replicates each having two or more positive events is 0.0955 x 0.0955 = 0.00911 (0.911%). A false positive rate of almost 1% is not acceptable for a clinical assay.

Changing the threshold from 2 to 3 events decreases the probability of false positive events (≥3) from a single reaction to 0.0157 and false positive in duplicate reactions to 0.000247 (0.02%).

For the exon 19 deletion assay, there were no false positives observed under experimental conditions. Applying a statistical rule of 3 (if an event does not occur in n observations, then the 95% confidence interval can be approximated by 3/n), changing the threshold from 2 to 3 events will decrease the false positive 95% confidence interval from 0.00304 (0.3%) to <0.001%.

You can rationalize setting the exon 19 threshold lower, but we have decided to keep it at 3 in our lab for convenience in clinical interpretation. There is value is having a single threshold for both assays to minimize risk of misinterpretation in a laboratory with many molecular pathologists rotating on service.

We have added the following line to the manuscript:

With a positive threshold set at 3 events in duplicate reactions, the expected false positive rate in EGFR L858R analysis was 0.02%.

5, there is a significant difference in terms of analytical and clinical sensitivity between the ddPCR assays and that reported previously such as ref. 15. This should be discussed with sufficiently deepth so that the readers can understand the reasons behind.

We performed additional analysis, reporting the demographic and clinical characteristics of patients in our cohort. These are addressed in detail in the response to reviewer #2. We noted that most patients in our cohort had been pretreated. Sensitivity among patients with Stage IV disease was 59% and for patients who had not received prior therapy was 71%.

During our reanalysis, we identified a data entry error. The final sensitivity for plasma ddPCR was 49/92 = 53%. It was erroneously reported as 48/93 in the prior version. All cases are shown in the minimal data set.

6, considering the broad application of NGS for ctDNA in clinic, it would be more informative if there is a head-to-head comparison between the ddPCR assays and a NGS assay for ctDNA detection.

We agree that a direct comparison between ddPCR assay and NGS assay for ctDNA detection would be informative. Unfortunately, our lab has not optimized our NGS assay to the depth of sequencing required for ctDNA detection, and this topic is beyond the scope of the current report.

Reviewer #2:

The manuscript is aimed at validating a dual channel ddPCR system for EGFR L858R and exon 19 deletion detection in plasma and FFPE tissue. Despite the analysis have been conducted in a very rigorous way, the number of mutant samples analysed on FFPE is very low (6 out of 20) and the advantages in terms of specificity and turn around time of a ddPCR are well known.

Thank you for your comment.

Considering the low sensitivity (only 52%) using plasma samples, I would suggest the authors to accurately consider/discuss the clinical characteristics of patients, since this is one of the major challenge of liquid biopsy. Patients should try to set up their assay on an homogeneous population, to cut out any possible analytical bias to confirm their assay works.

We have reviewed the clinical characteristics of our cohort and reexamined each case. During our reanalysis, we identified a data entry error. The final sensitivity for plasma ddPCR was 49/92 = 53%.

The clinical characteristics of the patients are included in the updated results and provided here:

In this cohort of patients, the median age was 64 years (range 31 to 97 years) with 64 (70%) female patients. New diagnoses accounted for 28 of 92 (30%) patients with EGFR-mutated cancers with 82 (89%) being at stage IV and 89 (97%) being at stage III or IV. Most patients had received some form of treatment prior to their first plasma ddPCR assay, with 52 (57%) receiving a tyrosine kinase inhibitor, 21 (23%) receiving chemotherapy, and 36 (39%) receiving localized therapy to their primary tumor either in the form of surgery (20, 22%) or radiotherapy (16, 17%). Only 28 (30%) had received no pre-testing therapy for their primary tumor at all.

Sensitivity among patients with Stage IV disease was 59%, while for patients with Stage I-II disease, it was only 17%. Sensitivity also varied by treatment received, with sensitivity for patients receiving no treatment being 71%, for those receiving a tyrosine kinase inhibitor being 44%, and for those receiving any systemic therapy (tyrosine kinase inhibitor, chemotherapy, or checkpoint inhibitor) being 37%.

We have also updated the discussion:

Our cohort has a slightly lower proportion of stage IV patients than most of those previously published. Additionally, more than half of our patients (56%) had already been treated with one or more tyrosine kinase inhibitors at the time of testing which may reduce the ability of our assay to detect mutations, as evidenced by the markedly decreased sensitivity of our assay for patients who has received systemic therapy (37%) versus those who had not received any treatment at all (71%). 

Thank you for taking time to review our manuscript. We appreciate your input in improving the article.

Sincerely,

Fei Dong, M.D.

Brigham and Women's Hospital

Boston, MA

---

## [Decision Letter · Decision Letter 1]

7 Feb 2022

Detection of EGFR mutations in non-small cell lung cancer by droplet digital PCR

PONE-D-21-21322R1

Dear Dr. Dong,

We’re pleased to inform you that your manuscript has been judged scientifically suitable for publication and will be formally accepted for publication once it meets all outstanding technical requirements.

Kind regards,

Alvaro Galli

Academic Editor

PLOS ONE

Additional Editor Comments (optional):

Reviewers' comments:

Reviewer's Responses to Questions

**Comments to the Author**

1. If the authors have adequately addressed your comments raised in a previous round of review and you feel that this manuscript is now acceptable for publication, you may indicate that here to bypass the “Comments to the Author” section, enter your conflict of interest statement in the “Confidential to Editor” section, and submit your "Accept" recommendation.

Reviewer #1: All comments have been addressed

Reviewer #2: All comments have been addressed

2. Is the manuscript technically sound, and do the data support the conclusions?

Reviewer #1: Yes

Reviewer #2: Yes

3. Has the statistical analysis been performed appropriately and rigorously? 

Reviewer #1: Yes

Reviewer #2: Yes

4. Have the authors made all data underlying the findings in their manuscript fully available?

Reviewer #1: Yes

Reviewer #2: Yes

5. Is the manuscript presented in an intelligible fashion and written in standard English?

Reviewer #1: Yes

Reviewer #2: Yes

6. Review Comments to the Author

Reviewer #1: All my previous comments have been well addressed. I have no further comments and am happy to see the manuscript to be accepted for publication on the journal.

Reviewer #2: (No Response)

7. PLOS authors have the option to publish the peer review history of their article (what does this mean?). If published, this will include your full peer review and any attached files.

Reviewer #1: No

Reviewer #2: No

---

## [Editor Report · Acceptance letter]

15 Feb 2022

PONE-D-21-21322R1 

Detection of EGFR mutations in non-small cell lung cancer by droplet digital PCR 

Dear Dr. Dong:

I'm pleased to inform you that your manuscript has been deemed suitable for publication in PLOS ONE. Congratulations! Your manuscript is now with our production department. 

Kind regards, 

on behalf of

Dr. Alvaro Galli 

Academic Editor

PLOS ONE